# Pigment Extracts of *Tetradesmus obliquus*, *Phaeodactylum tricornutum* and *Desmodesmus armatus* Exert Anti-Adipogenic Effects on Maturing 3T3-L1 Pre-Adipocytes

**DOI:** 10.3390/ijms262110314

**Published:** 2025-10-23

**Authors:** Helen Carr-Ugarte, Leixuri Aguirre, María P. Portillo, Gerardo Álvarez-Rivera, Sergio Seoane, Pablo Aramendi, Itziar Eseberri

**Affiliations:** 1Nutrition and Obesity Group, Department of Nutrition and Food Science, Lucio Lascaray Research Institute, University of the Basque Country (UPV/EHU), 01006 Vitoria-Gasteiz, Spain; helen.carr@ehu.eus (H.C.-U.); mariapuy.portillo@ehu.eus (M.P.P.); itziar.eseberri@ehu.eus (I.E.); 2CIBEROBN Physiopathology of Obesity and Nutrition, Institute of Health Carlos III, 01006 Vitoria-Gasteiz, Spain; 3Bioaraba Health Research Institute, 01006 Vitoria-Gasteiz, Spain; 4CRETUS, Department of Analytical Chemistry, Nutrition and Food Science, Universidade de Santiago de Compostela, 15782 Santiago de Compostela, Spain; gerardo.alvarez.rivera@usc.es; 5Department of Plant Biology and Ecology, University of the Basque Country (UPV/EHU), 48940 Leioa, Spain; sergio.seoane@ehu.eus (S.S.); pablo.aramendi@ehu.eus (P.A.); 6Research Centre for Experimental Marine Biology and Biotechnology (Pentzia Marine Station, PiE-UPV/EHU), 48620 Plentzia, Spain

**Keywords:** microalgae, *Tetradesmus obliquus*, *Phaeodactylum tricornutum*, *Desmodesmus armatus*, 3T3-L1, pre-adipocytes, adipogenesis, obesity

## Abstract

Microalgae have attracted the interest of researchers due to their high amounts of bioactive compounds with potential anti-obesity effects. In this context, the aim of this study is to analyse the effects of pigment extracts of *Tetradesmus obliquus*, *Phaeodactylum tricornutum* and *Desmodesmus armatus* on triglyceride accumulation in 3T3-L1 pre-adipocytes. Pigments were extracted and the chlorophyll and carotenoid profiles were analysed by HPLC-DAD-APCI-QTOF-MS analysis. Next, the three extracts were tested in maturing 3T3-L1 pre-adipocytes treated during 8 days at doses of 6.25, 12.5, 25 and 50 µg/mL. Cell viability was evaluated and triglyceride content of cells was measured by a commercial kit. Furthermore, adipogenic gene expression was measured in cells treated with the highest dose of the three extracts. The characterisation showed that the predominant pigments in each extract were different among the microalgae, with fucoxanthin being the main one in *Phaeodactylum tricornutum* and chlorophylls, lutein and violaxanthin/neoxanthin in the other two microalgae. All the tested microalgae extracts reduced triglyceride content of pre-adipocytes, although differing in the minimum effective dose. The underlying mechanism depends on the analysed extract, but the three extracts reduced adipogenesis via *Pparg* inhibition. In conclusion, the pigment extracts of the three microalgae exert anti-adipogenic effects in 3T3-L1 pre-adipocytes.

## 1. Introduction

Obesity represents a major public health challenge, currently affecting over one billion adults, adolescents and children. According to the World Health Organization (WHO), in 2022, 43% of adults were overweight and 16% were living with obesity [1]. Obesity is defined as a chronic, multifactorial disease that is characterised by excessive body fat accumulation. It is recognised as a risk factor for the development of various conditions, including type 2 diabetes, cardiovascular disease and certain types of cancer.

Diet and physical activity remain the primary strategies for the prevention and treatment of this disease. There is an increasing interest in the development of novel therapies based on bioactive molecules present in foodstuffs and plants, used as dietary functional ingredients. In this context, many researchers have focused on macroalgae and microalgae, which are rich in several bioactive compounds such as pigments, polyunsaturated fatty acids, phenolic compounds, peptides, lipids, vitamins, polysaccharides and sterols. Microalgae have been reported to exert positive effects in the context of obesity and to exhibit anti-diabetic, hypolipidemic, anti-hypertensive and anti-cancer activities [2], presenting certain advantages over macroalgae, such as a higher growth rate [3], or a high standardization in production under controlled conditions, which allows producing high concentrations of key nutrient [4].

The selection of the three microalgae strains was based on the fact that we were interested in analysing the effects of microalgae belonging to different classes (*Chlorophyceae* and *Bacillariophyceae*), and also in testing potential differences in the response to extracts from microalgae of the same class (*Desmodesmus armatus* and *Tetradesmus obliquus). Tetradesmus obliquus* and *Desmodesmus armatus* are green microalgae, which present high contents of chlorophylls, lutein and zeaxanthin. *Phaeodactylum tricornutum* is a yellow-brown microalga, whose main pigment is fucoxanthin [5].

The anti-obesity effect of *Phaeodactylum tricornutum* and its main pigments have been demonstrated in in vitro studies [6,7]. In the case of *Tetradesmus obliquus* and *Desmodesmus armatus,* although no studies have been published yet, the anti-obesity effects of its main pigments, chlorophyll, lutein and zeaxanthin, have been reported [8,9,10,11].

The main objective of the present study is to analyse the effects of pigment extracts, obtained from three microalgae, *Tetradesmus obliquus*, *Phaeodactylum tricornutum* and *Desmodesmus armatus*, on triglyceride accumulation in 3T3-L1 pre-adipocytes. In addition, the study explores the underlying mechanisms responsible for the observed effects.

## 2. Results

### 2.1. Characterisation of Microalgae Extracts

The pigments identified in the extracts of *Tetradesmus obliquus*, *Phaeodactylum tricornutum* and *Desmodesmus armatus* are presented in Table 1, whereas the relative abundance profiles are displayed in Figure 1. A total of 18 major pigments were tentatively identified based on their maximum absorption wavelength (λmax), molecular ion (*m*/*z*), and characteristic MS/MS fragments obtained through LC-DAD-APCI(+)-QTOF-MS/MS analysis. The total wavelength UHPLC-DAD profile revealed a high abundance of carotenoids and chlorophylls (Figure 2), confirming species-specific pigment signatures: a fucoxanthin- and diadinoxanthin-rich profile in the diatom *Phaeodactylum tricornutum*, contrasted with lutein- and violaxanthin/neoxanthin-dominated profiles in the chlorophytes *Tetradesmus obliquus* and *Desmodesmus armatus*.

Carotenoids were assigned by the presence of three characteristic absorption maxima between 400 and 475 nm, their molecular ions, and fragment ion patterns. Chlorophyll-related pigments were recognised by their Soret-band absorption maxima at 400–450 nm and accurate mass values matching known porphyrins. Chlorophyll assignments were supported by diagnostic fragments and corresponding to losses of the phytyl chain.

### 2.2. Effects on Cell Viability and Triglyceride Content

After cell incubation with extracts of *Tetradesmus obliquus*, *Phaeodactylum tricornutum* and *Desmodesmus armatus* at doses of 6.25, 12.5, 25 or 50 µg/mL from day 0 to day 8 of differentiation (adipogenesis stage), no cytotoxic effects were observed in the cells (Figure 3).

All the extracts reduced triglyceride accumulation during the adipogenic process of the cells (Figure 4). *Desmodesmus armatus* extract decreased triglyceride content at all tested doses, with reduction percentages of 18%, 22%, 32% and 38%, respectively. *Tetradesmus obliquus* extract was effective at 12.5 (41%), 25 (37%) and 50 µg/mL (52%). In *Phaeodactylum tricornutum*, only the highest dose (50 µg/mL) reduced lipid accumulation, with triglyceride content reduced by 46%.

### 2.3. Effect of Microalgae Pigment Extracts on Gene Expression in Maturing Pre-Adipocytes

To elucidate the mechanisms underlying the observed effects on lipid accumulation in maturing pre-adipocytes, the expression of genes involved in adipogenesis was analysed in cells treated with the highest dose (50 µg/mL) of each extract (Figure 5).

Treatment of pre-adipocytes with the three extracts caused a reduction in peroxisome proliferator-activated receptor γ (*Pparg*) mRNA levels. In cells treated with *Tetradesmus obliquus*, the other genes analysed remained unchanged. *Phaeodactylum tricornutum* also reduced mRNA levels of CCAAT/enhancer-binding protein α (*Cebpa)*, fatty acid synthase *(Fasn)* and acetyl-CoA carboxylase (*Acc*). Pre-adipocytes treated with the *Desmodesmus armatus* extract additionally showed a down-regulation of *Fasn* and an up-regulation of sterol regulatory element-binding transcription factor 1 (*Srebf1)*. None of the extracts induced changes in genes related to the initial stage of adipogenesis (*Creb1* and *Klf5*).

## 3. Discussion

As explained in the Introduction, microalgae are being extensively studied due to their potential applications in the nutraceutical and pharmaceutical industries, among others. They are notable for their high nutritional value and serve as a rich source of bioactive compounds, including chlorophylls and carotenoids. These compounds have demonstrated considerable potential in the prevention and management of several chronic diseases, such as cardiovascular conditions, cancer, diabetes and obesity [12,13,14]. In the context of obesity, observational studies in humans have reported an inverse correlation between serum carotenoid levels and body mass index, body fat accumulation and metabolic syndrome [15,16]. Moreover, supplementation with these compounds has been associated with anti-obesity effects [17].

Nevertheless, few studies have investigated the potential anti-obesity effects of microalgae extract and the underlying mechanisms involved. In this scenario, the present study evaluated pigment extracts from *Tetradesmus obliquus*, *Phaeodactylum tricornutum* and *Desmodesmus armatus* in 3T3-L1 maturing pre-adipocytes to assess their impact on adipogenesis. This process, which involves the formation of mature adipocytes from precursor cells, plays a key role in the expansion of adipose tissue, and thus in the development of obesity and its associated co-morbidities [18]. As indicated in the Introduction, we were interested in analysing the effects of microalgae belonging to different classes (Chlorophyceae and Bacillariophyceae), and also in testing for potential differences in the response to extracts from microalgae of the same class (*Desmodesmus armatus* and *Tetradesmus obliquus).*

The relative pigment distribution (Figure 1) revealed clear differences between *Phaeodactylum tricornutum* and the green microalgae. In *Phaeodactylum tricornutum*, fucoxanthin accounted for the largest fraction of the carotenoid pool, consistent with its role as the major light-harvesting xanthophyll in diatoms [19]. Diadinoxanthin and diatoxanthin were also present at relevant proportions, evidencing the activation of the diadinoxanthin–diatoxanthin cycle, which is well documented in diatoms as a central photoprotective pathway [20].

In contrast, the green microalgae *Tetradesmus obliquus* and *Desmodesmus armatus* exhibited pigment profiles dominated, among carotenoids, by lutein and the carotenoids of violaxanthin-cycle. This pattern is consistent with previous reports on chlorophytes, where lutein is the principal pigment and the violaxanthin–antheraxanthin–zeaxanthin cycle operates as the main photoprotective mechanism [21].

Interestingly, β-carotene was not detected in any of the three microalgal extracts. This may be explained by several factors. For instance, in diatoms, fucoxanthin and diadinoxanthin largely replace β-carotene as accessory pigments in light-harvesting complexes [22]. It cannot be excluded that part of the β-carotene was consumed during stress responses or converted into secondary apocarotenoids. Therefore, the absence of β-carotene in the current profiles most likely reflects its naturally low levels and functional replacement by other xanthophylls in these microalgae, not excluding other reasons like potential extraction and analytical limitations.

Chlorophyll a was the most abundant porphyrin in all three species, while chlorophyll b and its derivative were consistently detected in the chlorophytes *Tetradesmus obliquus* and *Desmodesmus armatus*, as expected for their taxonomic group [23]. In contrast, *Phaeodactylum tricornutum* lacked chlorophyll b, but contained significant amounts of fucoxanthin and xanthophyll-cycle pigments, reinforcing the diatom-specific light-harvesting apparatus [19]. Minor amounts of pheophorbide and pheophytin derivatives were observed across species, suggesting partial degradation or demetalation of chlorophylls, probably during extraction.

Concerning the cell culture experiments addressed in the present study, it can be pointed out that one of the advantages of in vitro studies is the possibility of performing dose screening. In the present study, the selected concentrations for evaluating the effects of microalgae extracts on adipogenesis were 6.25, 12.5, 25 and 50 µg/mL. These concentrations were chosen based on two criteria. First, the selected range (in µg/mL) was commonly used in previous studies in the field [2,24]. Second, the upper limit of 50 µg/mL was established to ensure that the pigment concentrations in the extracts were comparable to those potentially found in plasma and tissues following supplementation in animal models [25].

The three extracts effectively reduced triglyceride accumulation in pre-adipocytes. The absence of any decrease in cell viability suggests that these effects are attributable to anti-adipogenic activity. Among them, *Desmodesmus armatus* extract appeared to be the most influential, as it significantly reduced triglyceride accumulation even at the lowest tested dose (6.25 µg/mL). However, the overall percentages of reduction induced by this extract were lower than those observed with the other two.

It is noteworthy that both green microalgae reduced triglyceride accumulation across a wide range of doses, whereas only the highest dose of *Phaeodactylum tricornutum* produced a comparable effect. Based on the pigment profile, it can be hypothesised that chlorophyll, which is more abundant in *Tetradesmus obliquus* and *Desmodesmus armatus* than in *Phaeodactylum tricornutum*, may exert a stronger anti-adipogenic effect than fucoxanthin, which is exclusive to *Phaeodactylum tricornutum.* Although fucoxanthin has shown anti-obesity effects in previous studies, a higher amount of this carotenoid might be required to achieve similar effects in this model. Another possibility could be the different carotenoid profiles present in chlorophyceae and diatoms. Nevertheless, more research, based on the incubation of cells with the isolated compounds and different bioactive compound fractions, is needed in order to confirm this hypothesis.

To date, few studies have investigate the anti-adipogenic effect of microalgae extracts in 3T3-L1 maturing pre-adipocytes, which limits direct comparison between the present results and those available in the literature. Among the microalgae tested, only studies involving *Phaeodactylum tricornutum* have been published. Koo et al. reported that the incubation of 3T3-L1 pre-adipocytes with an extract of this microalga reduced triglyceride accumulation at a dose (250 µg/mL) considerably higher than those applied in the present study [26]. In addition, the effects of fucoxanthin, the main carotenoid of this microalga, on adipogenesis have been addressed. One study reported that fucoxanthin reduced triglyceride accumulation in 3T3-L1 pre-adipocytes at a dose of 40 µM. Other studies have also demonstrated that fucoxanthin suppresses adipogenesis in 3T3-L1 pre-adipocytes at concentrations as low as 10 µM [6,7,27]. Furthermore, lutein and zeaxanthin have also shown anti-adipogenic and lipid-reducing effects in several pre-clinical studies, suggesting that they could be responsible, at least in part, for the triglyceride reduction observed in pre-adipocytes treated with both *Tetradesmus obliquus* and *Desmodesmus armatus* [9,10,11].

Anti-obesity effects of chlorophylls have also been described in previous studies. For instance, Seo et al. showed that chlorophyll a at 5 and 10 µg/mL inhibited the differentiation of 3T3-L1 pre-adipocytes [28]. Another study reported a marked reduction in intracellular lipid accumulation in 3T3-L1 maturing pre-adipocytes following treatment with a chlorophyll derivative [8]. In addition, studies conducted in rodents showed that supplementation with chlorophyll and chlorophyll-rich extracts helped prevent weight gain and obesity-associated pathologies [29,30]. The results of the present study are consistent with previously reported data. Pheophorbide is a derivative product of chlorophyll breakdown. Some authors have reported that this chlorophyll derivative inhibits the adipogenesis via the reduction in lipid accumulation [31,32]. Park et al. reported that pheophorbide reduced lipid content in pre-adipocytes at doses of 17, 42 and 83 µM and reduced the adipogenesis by down-regulating the adipogenic transcription factors such as PPARg and C/EBPa [31]. In another study carried out by Liu et al., pre-adipocytes treated with 5, 10, and 20 μmol L^−1^ doses of pheophorbide also showed anti-adipogenic effects [32].

As the three microalgae extracts reduced lipid content in maturing pre-adipocytes, underlying mechanisms were investigated. Gene expression of key adipogenic markers was analysed via RT-PCR in cells treated at the highest extract concentration, which proved effective across all microalgae. Transcriptional regulation of this process is multifaceted: intracellular cAMP levels activate *Creb*, which in turn stimulates the initial phase of adipogenesis by up-regulating *Cebpb* and *Cebpd*. This is followed by expression of other adipogenic factors, notably *Pparg* -the master regulator- and *Cebpa*.

In the present study, treatment of maturing pre-adipocytes with the three extracts did not affect the initial phase of adipogenesis but reduced *Pparg* gene expression, which appears to be primarily responsible for the observed lipid reduction. As previously mentioned, this transcriptional regulator plays a pivotal role in the development of mature adipocytes. Indeed, some studies have indicated that *Pparg* is both necessary and sufficient to initiate adipogenesis [33,34], although usually *Pparg* and *Cebpa* cross-regulate each other [35]. In fact, a reduction in the expression of this transcription factor was also observed following treatment with *Phaeodactylum tricornutum*. Surprisingly, pre-adipocyte treatment with *Desmodesmus armatus* led to an increase in *Srebf1* mRNA levels, although this up-regulation did not result in enhanced *Pparg* expression or adipogenesis stimulation [36].

*Acc* and *Fasn* are involved in *de novo* lipogenesis and the resulting triglyceride accumulation. These genes are expressed during the later stages of cell differentiation. Their expression was also measured in response to extracts obtained from the three microalgae, and we observed that *Phaeodactylum tricornutum* and *Desmodesmus armatus* reduced *Fasn* expression. Additionally, *Phaeodactylum tricornutum* decreased *Acc* mRNA levels. In contrast, *Tetradesmus obliquus* did not induce any significant change in the expression of either genes. This fact suggests that the lipid reduction observed by *Tetradesmus obliquus* extract at 50 µg/mL is exclusively caused by *Pparg* inhibition.

The present results are consistent with those published by Koo et al. [26], who reported that incubation of 3T3-L1 pre-adipocytes with *Phaeodactylum tricornutum* extract significantly reduced *Pparg* protein expression, without changes in *Cebpa*. In the same study, dietary supplementation of 8-week-old female mice with up to 3.25 mg/kg body weight/day for six weeks also led to reduced *Pparg* protein expression in white adipose tissue. In this case, however, *Cebpa* protein expression was also decreased.

Regarding studies using fucoxanthin extracts, the authors reported effects similar to those found in the present study with the microalgal extract [26]. A reduction in *Pparg* expression, without changes in *Cebpa*, was observed when 3T3-L1 pre-adipocytes were incubated with 40 μM of the compound. Similarly, Maeda et al. found reduced Pparg protein expression after treating 3T3-L1 pre-adipocytes with low doses of fucoxanthin (2.5 and 5 μM) [6]. In other studies, both *Pparg* and *Cebpa* were reduced following incubation with fucoxanthin at doses as low as 3.125, 6.25 and 12.5 µM [7]. Thus, these data support the proposal that, although *Pparg* down-regulation is necessary for inhibition of adipogenesis, decreased *Cebpa* expression is not strictly required [33,34].

These results support the conclusion that pigment extracts from *Tetradesmus obliquus*, *Phaeodactylum tricornutum* and *Desmodesmus armatus* exert anti-adipogenic effects, with the green microalgae demonstrating greater influence at lower doses. This effect appears to be mediated by the down-regulation of *Pparg*, the master regulator of this process. In the present study, gene silencing/overexpression experiments, that would have provided a more strong evidence of the involvement of *Pparg* in the reduction in adipogenesis, have not been addressed. Consequently, this represents a limitation of the study.

Although compounds that reduce adipogenesis have traditionally been considered potential anti-obesity agents [37,38], adipogenesis has recently emerged as a therapeutic strategy to enhance adipose tissue health and counteract metabolic complications associated with adipocyte hypertrophy. Therefore, considering this fact, and taking into account that in in vitro studies important aspects, such as crosstalk among tissues and bioactive compound bioavailability, cannot be assessed, further studies in animal models are needed to explore the anti-obesity potential of these microalgae extracts, as well as their effects on obesity-related co-morbidities such as insulin resistance and dyslipidemia. Moreover, in order to be sure if these extracts are useful for clinical uses, intervention studies in humans are also needed in a final research step.

## 4. Materials and Methods

### 4.1. Microalgae Production

Three microalgae were selected for this study: *Tetradesmus obliquus* (BMCC738), *Phaeodactylum tricornutum* (RCC69; GenBank accession number KT860965) and *Desmodesmus armatus* (BMCC598). *Tetradesmus obliquus* and *Desmodesmus armatus* are green microalgae belonging to the *Chlorophyceae*, whereas *Phaeodactylum tricornutum* is a yellow-brown microalga from the *Bacillariophyceae.* All strains were provided by the Basque Microalgae Culture Collection (BMCC) at the University of the Basque Country (UPV/EHU). Inocula were maintained in 10 mL tubes at 17 ± 2 °C under a 12:12 h light:dark cycle. Illumination consisted of white light, combining cold and warm fluorescent tubes, with an intensity of approximately 60 µEm^−2^s^−1^ measured at the culture tube surface. The culture medium was prepared using natural, filtered (0.22 µm) freshwater or seawater depending on the species, according to the F/2 recipe [39].

The cultures were grown photoautotrophically in 100 L bubble column PBRs (Figure 6). The PBR consisted of a clear methacrylate column with an internal diameter of 250 mm and a height of 2 m. To mix the culture broth, sterile filtered air was bubbled through the vessel at a flow rate of 6 to 8 L/min. Illumination was provided by 35W VALOYA LED tubes (Valoya, Helsinki, Finland) with the NS12 spectrum, delivering an intensity of 250 µEm^−2^s^−1^ measured at the midpoint of the empty column. A 16:8 h light:dark cycle was maintained, and the temperature was controlled at 19 ± 2 °C. The pH was regulated at pH 8 through automatic injecting of carbon dioxide as needed. The fresh culture medium was prepared following the same protocol as for the inocula.

The culture growth was measured by the optical density at 680 nm (OD_680_) in a spectrophotometer. Two days after the cultures reached the stationary phase, samples were collected and centrifuged in discontinuous mode using a GEA Westfalia OTC3 (GEA group, Düsseldorf, Germany) at 1197 g during 15 min. The resulting biomass was then stored at −80 °C until the analysis.

### 4.2. Microalgae Extraction Preparation

To determine the optimal conditions for pigment extraction from microalgae, various solvents at different proportions were tested. A mixture of ethanol:water (70:30 *v*/*v*) was selected as the most effective, and pigment extracts from the three microalgae were obtained using a modified version of the method described by Cesario et al. [40]. Specifically, 1 g of freeze-dried microalgal biomass was dissolved in 20 mL of the ethanol:water solution and sonicated with a Branson Sonifier SFX550 (Emerson, San Luis, MO, USA) at 40% amplitude for 4 min. After sonication, the samples were kept for 12 h in an orbital shaker at 4 °C. Samples were then centrifuged at 12,000× *g* for 10 min at 4 °C, and the supernatant was filtered using two nanopore filters. Lastly, ethanol was evaporated under N_2_ steam, and the pigment extracts were freeze-dried and stored at −80 °C until use in cell treatments.

### 4.3. Determination of Carotenoid and Chlorophyll Profiles by LC-DAD-QTOF-MS/MS

Freeze-dried extracts were resuspended in ethanol:water (70:30 v:v) and subsequently diluted in methanol to an appropriate concentration (1–10 mg mL^−1^) for analysis. Prior to assessment, samples were centrifugated at 14,000 rpm to remove insoluble impurities.

Carotenoid and chlorophyll profiling was performed by HPLC-DAD-APCI-QTOF-MS/MS on an Agilent 1290 UHPLC system (ultrahigh-performance liquid chromatography) (Agilent, Santa Clara, CA, USA) equipped with a diode-array detector (DAD) (Agilent, Santa Clara, CA, USA) and coupled to an Agilent 6540 quadrupole time-of-flight mass spectrometer (q-TOF MS) (Agilent, Santa Clara, CA, USA) with an atmospheric pressure chemical ionisation (APCI) source. Chromatographic separation was performed on a Thermo Fisher Scientific Accucore C30 column (2.6 μm, 4.6 × 50 mm) (Thermo Fisher, Waltham, MA, USA) maintained at 30 °C. The mobile phases consisted of methanol–MTBE–water (90:7:3 v/v/v) as solvent A and methanol–MTBE (10:90 *v*/*v*) as solvent B. The elution program was as follows: 0 min, 0% B; 9.5 min, 40% B; 11 min, 80% B; 12 min, 100% B; 13.5 min, 100% B; 15 min, 0% B. The injection volume was 5 µL and the flow rate, 0.8 mL/min.

The mass spectrometer was operated in positive ionisation mode (APCI+) with the following settings: gas temperature, 300 °C; drying gas, 8 L/min; vaporiser temperature, 350 °C; nebuliser pressure, 40 Psi; capillary voltage, 3500 V; corona, +4 μA; fragmentor voltage, 110 V; and skimmer voltage, 45 V. Both MS and tandem MS modes were used for structural elucidation of target phytochemicals. The MS and auto MS/MS modes were configured to acquire m/z values in the range of 25–1500, at a scan rate of 10 spectra per second. Auto MS/MS mode was operated using two collision-induced dissociation energies (20 and 40 eV), selecting four precursor ions per cycle at a threshold of 200 counts.

A DAD system was used to monitor carotenoid and pigment profiles across a wavelength range of 190–640 nm (peak width: 0.1 min (2 s), slit width: 4 nm). To obtain complementary structural information, ultraviolet–visible (UV–VIS) profiles and QTOF-MS/MS data acquired in positive ionisation mode using an atmospheric pressure chemical ionisation (APCI) source were jointly analysed.

Peak area values (%) from the LC-DAD chromatograms were reported for each pigment as an estimate of its relative abundance, in order to comparatively evaluate the pigment profiles of the three target microalgae extracts. Due to the tentative annotation of some carotenoid isomers, this approach allowed us to compare extracts with similar profile in the absence of standards.

### 4.4. Experimental Design

3T3-L1 pre-adipocytes were obtained from the American Type Culture Collection (Manassas, VA, USA) and cultured in Dulbecco’s Modified Eagle Medium (DMEM) (Sigma-Aldrich, Madrid, Spain) supplemented with 10% fetal bovine serum (FBS) and 1% penicillin/streptomycin (PS; 10,000 U/mL). Two days after reaching confluence, pre-adipocytes were induced to differentiate using a medium composed of DMEM with 10% FBS and 1% PS, 10 µg/mL insulin, 0.5 mM isobutylmethylxanthine (IBMX) and 1 mM dexamethasone, and incubated for two days. After this period, cells were preserved in FBS/DMEM/PS supplemented with 10 µg/mL insulin for two additional days. From day four onward, the insulin concentration was reduced to 0.2 µg/mL in the incubation medium. All cultures were maintained at 37 °C in a humidified atmosphere with 5% CO_2_ throughout the experimental period.

### 4.5. Cell Treatments

Cells were incubated with microalgae extracts (*Tetradesmus obliquus*, *Phaeodactylum tricornutum* and *Desmodesmus armatus*) at concentrations of 6.25, 12.5, 25 or 50 µg/mL (diluted in methanol) during the adipogenic process (day 0–day 8). The incubation medium containing either the extracts or the vehicle was refreshed on days 0, 2, 4 and 6. On day 8 of differentiation, cells were harvested for TG and protein quantification, and RNA was extracted for gene expression analysis. Each experiment was conducted in triplicate.

### 4.6. Cell Viability Assay

Cell viability was assessed using the crystal violet assay, which is based on DNA staining of viable seeded cells [41]. Subsequent to the treatments described above, cells were washed with phosphate-buffered saline (PBS), fixed with 3.7% formaldehyde, and stained with 0.25% crystal violet in the dark for 20 min. The resulting crystals were then solubilised with 33% acetic acid, and absorbance was recorded at 590 nm using an iMark microplate reader (Bio-Rad, Hercules, CA, USA). The absorbance values were directly proportional to cell density, and results are expressed in arbitrary units.

### 4.7. Determination of Triglyceride Content

Following cell treatments, the medium was removed, and adipocytes were thoroughly washed with PBS. Cells were then harvested in 300 µL of buffer comprising Tris-HCl, 150 mM NaCl and 1 mM EDTA supplemented with protease inhibitors. Cell disruption was carried out using a Branson Digital Sonifier SFX 550 (Emerson Electric Co., St. Louis, MO, USA) equipped with a 2 mm ultrasound microprobe (Biogen Scientific S.L., Madrid, Spain). TG content was quantified using a commercial kit (Spinreact, Girona, Spain), and protein concentration was determined by the Bradford et al. method [42] to standardise lipid content. Results are expressed as mg of triglycerides per mg of protein and presented in arbitrary units.

### 4.8. Analysis of Gene Expression by Real-Time PCR

RNA was extracted from cells using 700 µL of TRIzol (Invitrogen, Carlsbad, CA, USA) per well, following the manufacturer’s instructions. RNA integrity was assessed with the RNA 6000 Nano Assay (Thermo Scientific, Wilmington, DE, USA). Samples were subsequently treated with DNase I (Applied Biosystems, Foster City, CA, USA) to remove genomic DNA contamination. For complementary DNA (cDNA) synthesis, 1.75 µg of RNA from each sample was reverse-transcribed using the iScript cDNA Synthesis Kit (Bio-Rad, Hercules, CA, USA) under the following thermal conditions: 25 °C for 10 min, followed by incubation at 37 °C for 120 min, and completed at 85 °C for 5 min.

Relative mRNA levels of *Acc*, *Cebpa*, *Creb1*, *Fasn*, *Klf5*, *Pparg* and *Srebf1* were quantified using an iCycler-MyiQ Real-Time PCR Detection System (Bio-Rad, Hercules, CA, USA). β-actin mRNA levels were also measured and used as the reference gene. A total of 4.75 μL of each diluted cDNA sample was added to the PCR reagent mixture, which contained SYBR Green Master Mix (Applied Biosystems, Foster City, CA, USA), and upstream and downstream primers at a concentration of 5 µM. Primer sequences are provided in Table 2. Results are expressed as fold changes in threshold cycle (Ct) values relative to the control, calculated using the 2^−ΔΔCt^ method [43].

### 4.9. Statistical Analysis

The results were analysed using SPSS Statistics software 26.0 (SPSS, Chicago, IL, USA) and are presented as mean ± standard error of the mean (SEM). Comparisons between control cells and those treated with each microalgae extract were made using Student’s *t* test. Statistical significance was defined as *p* ≤ 0.05.

## Figures and Tables

**Figure 1 ijms-26-10314-f001:**
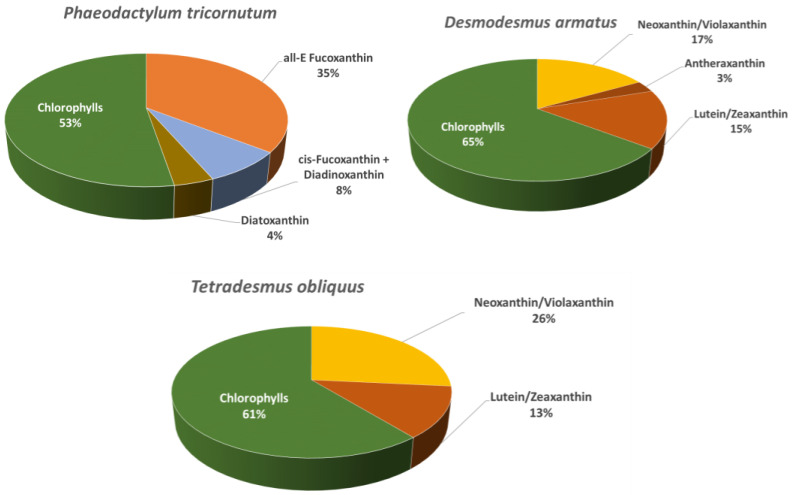
Relative abundance (area %) of the tentatively identified carotenoids and chlorophylls in the extracts of the three microalgae.

**Figure 2 ijms-26-10314-f002:**
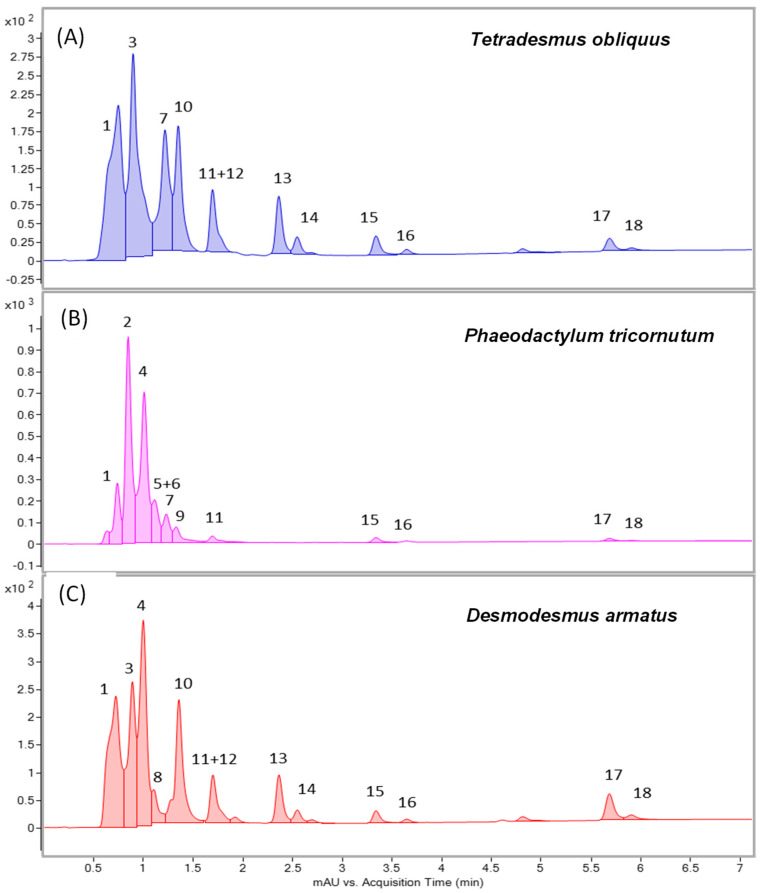
Full wavelength HPLC-DAD chromatograms (TWC, 190-640 nm) of *Tetradesmus obliquus* (**A**), *Phaeodactylum tricornutum* (**B**) and *Desmodesmus armatus* (**C**) extracts. Peak annotations are provided in Table 1.

**Figure 3 ijms-26-10314-f003:**
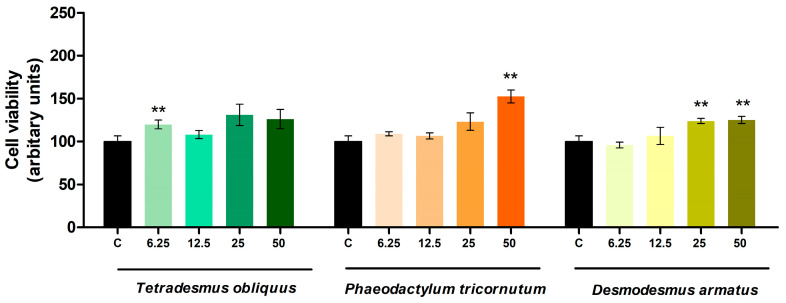
Cell viability of 3T3-L1 maturing pre-adipocytes treated with 6.25, 12.5, 25 or 50 μg/mL of *Tetradesmus obliquus*, *Phaeodactylum tricornutum* and *Desmodesmus armatus* microalgae extracts. Control cells (C) were not treated with microalgae extracts. Values are presented as means ± SEM. Comparisons between each dose and the control were analysed using Student’s *t*-test. Asterisks indicate significant differences compared to controls (** *p* < 0.01).

**Figure 4 ijms-26-10314-f004:**
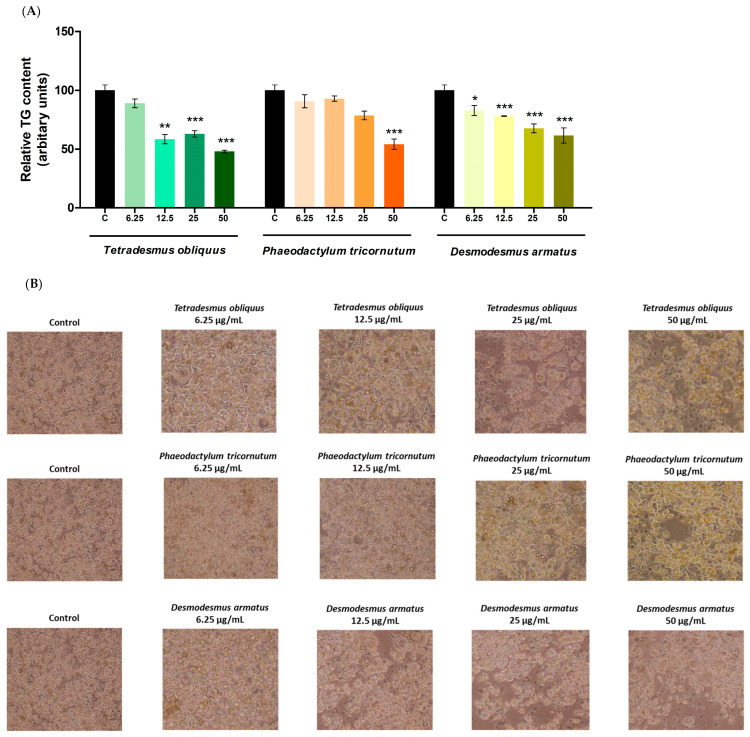
Effects of *Tetradesmus obliquus*, *Phaeodactylum tricornutum* and *Desmodesmus armatus* microalgae extracts at 6.25, 12.5, 25 and 50 µg/mL on triglyceride content in the adipogenic process of 3T3-L1 cells treated from day 0 to day 8 (**A**). Optical microscopy images show lipid accumulation at day 8 (**B**). Values are presented as means ± SEM. Comparisons between each dose and the control were analysed using Student’s *t*-test. Asterisks indicate significant differences compared to controls (* *p* < 0.05; ** *p* < 0.01; *** *p* < 0.001). TG: triglyceride.

**Figure 5 ijms-26-10314-f005:**
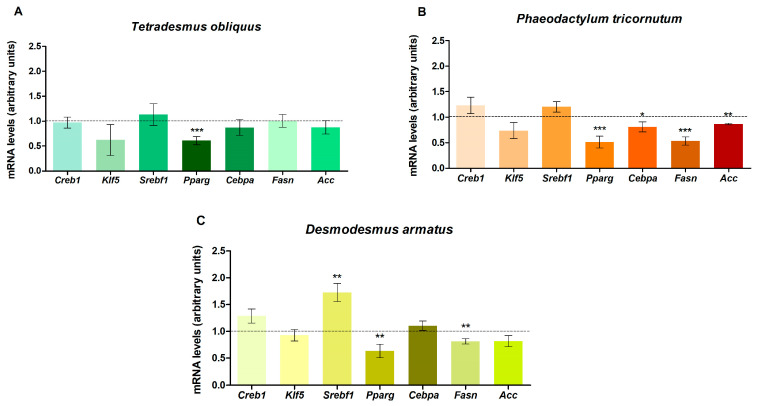
Effects of 50 µg/mL of *Tetradesmus obliquus* (**A**), *Phaeodactylum tricornutum* (**B**) and *Desmodesmus armatus* (**C**) microalgae extracts on gene expression of *Acc*, *Cebpa*, *Creb1*, *Fasn*, *Klf5*, *Ppara* and *Srebf1* in 3T3-L1 maturing pre-adipocytes treated from day 0 to day 8. Values are presented as means ± SEM. Comparisons between each microalgae and the control were analysed using Student’s *t*-test. Asterisks indicate significant differences compared to controls (* *p* < 0.05; ** *p* < 0.01; *** *p* < 0.001). Dashed lines represent values from the control cells. *Acc*: acetyl-CoA carboxylase; *Cebpa*: CCAAT/enhancer-binding protein α; *Creb1*: cAMP responsive element-binding protein 1; *Fasn*: fatty acid synthase; *Klf5*: Kruppel-like factor 5; *Pparg*: peroxisome proliferator-activated receptor γ; *Srebf1*: sterol regulatory element-binding transcription factor 1.

**Figure 6 ijms-26-10314-f006:**
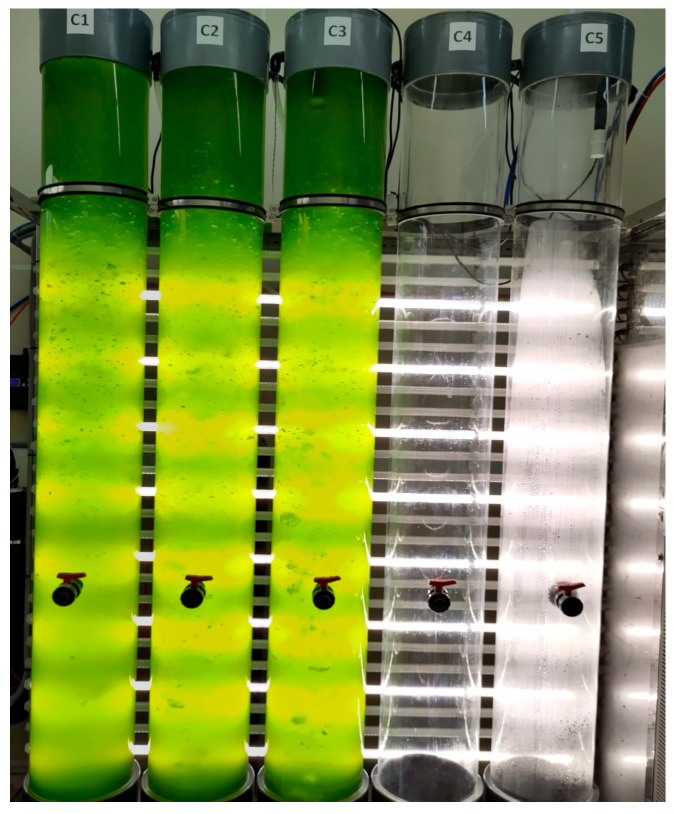
Cultivation of *Desmodesmus armatus* microalgae in bubble column PBRs.

**Table 1 ijms-26-10314-t001:** Tentatively identified carotenoids and chlorophylls in the extracts of the three microalgae, as determined by HPLC-DAD-APCI-QTOF-MS analysis, including peak annotations, high-resolution mass spectrometry data, and UV-Vis absorption maxima.

Peak No	RT (min)	Identification	Molecular Formula	m/z [M + H]^+^ (Theoretical)	Error (ppm)	MS/MS Product Ions	UV–Vis Maxima (nm)
1	0.724	Chlorophyll	-		-	-	400, 435, 470
2	0.847	all-E Fucoxanthin	C42H58O6	659.4306	2.6	641, 623, 581, 109	420 s, 445, 466 s
3	0.891	Neoxanthin-/Violaxanthin	C40H56O4	601.4251	2.8	583, 565, 167,116	316 s, 436, 464
4	0.998	Pheophorbide a	C35H36N4O5	593.2759	2.6	533, 505	410
5	1.0891	cis-Fucoxanthin	C42H58O6	659.4306	1.3	641, 623, 581, 109	330, 420 s, 445, 470 s
6	1.120	Diadinoxanthin	C40H54O3	583.4145	−0.8	565, 527, 283	424 s, 445, 474 s
7	1.214	Pheophorbide a-like	-		-	-	410
8	1.271	Antheraxanthin	C40H56O3	585.4302	−0.4	567, 493, 337, 231	380 s, 410, 455 s
9	1.333	Diatoxanthin	C40H54O2	567.4196	3.1	285, 217, 119	424 s, 450, 476
10	1.353	Lutein/Zeaxanthin	C40H56O2	569.4353	2.8	551, 337, 251	420 s, 444, 472
11	1.693	Chlorophyll	-		-	-	450
12	1.698	Echinenone	C40H54O	551.4247	0.3	533, 429, 175, 121	410
15	2.360	Chlorophyll b’	C55H70MgN4O6	907.5219	1.8	567, 476, 133	417, 440, 467
16	2.544	Chlorophyll b	C55H70MgN4O6	907.5219	2.4	583, 333, 167	417, 440, 467
17	3.338	Chlorophyll a’	C55H72MgN4O5	893.5426	4.5	593, 533	408
18	3.644	Chlorophyll a	C55H72MgN4O5	893.5426	4.5	593, 533	408
19	5.684	Pheophytin a’	C55H74N4O5	871.5732	−2.3	615, 583, 555	339, 390 s, 420 s, 432
20	5.904	Pheophytin a	C55H74N4O5	871.5732	−1.4	615, 583, 555	339, 390 s, 420 s, 432

**Table 2 ijms-26-10314-t002:** Primer sequences for quantitative real-time PCR amplification.

Gene	Sense Primer 5′-3′	Antisense Primer 5′-3′
*Acc*	GGA CCA CTG CAT GGA ATG TTA A	TGA GTG ACT GCC GAA ACA TCT C
*Cebpa*	TGG ACA AGA ACA GCA ACG AG	TCA CTG GTC AAC TCC AGC AC
*Creb1*	TTT GTC CTT GCT TTC CGA AT	CAC TTT GGC TGG ACA TCT TG
*Fasn*	AGC CCC TCA AGT GCA CAG TG	TGC CAA TGT GTT TTC CCT GA
*Klf5*	GGT CCA GAC AAG ATG TGA AAT GG	TTT ATG CTC TGA AAT TAT CGG AAC TG
*Pparg*	TCG CTG ATG CAC TGC CTA TG	GAG AGG TCC ACA GAG CTG ATT
*Srebf1*	GCT GTT GGC ATC CTG CTA TC	TAG CTG GAA GTG ACG GTG GT

*Acc*: acetyl-CoA carboxylase, *Cebpa*: CCAAT/enhancer-binding protein α, *Creb1*: cAMP responsive element-binding protein 1, *Fasn*: fatty acid synthase, *Klf5*: Kruppel-like factor 5, *Pparg*: peroxisome proliferator-activated receptor γ and *Srebf1*: sterol regulatory element-binding transcription factor 1.

## Data Availability

The original contributions presented in this study are included in the article. Further inquiries can be directed to the corresponding author(s).

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
