# Peer review of "Pigment Extracts of Tetradesmus obliquus, Phaeodactylum tricornutum and Desmodesmus armatus Exert Anti-Adipogenic Effects on Maturing 3T3-L1 Pre-Adipocytes"

_ijms, 2025, doi:10.3390/ijms262110314_

Round 1

Reviewer 1 Report

Comments and Suggestions for Authors

The manuscript deals with microalgae extracts derived after cultivation of these in bubble columns and consists of two major parts, the LC-MS-analysis of the occurring carotenoids and the effects of algae extracts on cell cultures (pre-adipocytes).

The overall impression on thoroughness on the style and form of the manuscript is low (e.g. wrong numbering of figure 6, line 189 vs. line 172, y-axis of figure 2 missing), line 404: 16:18 h?; line 347: 50 µM (based on what molecule/molecular weight?)

  • the introduction does not contain information on carotenoid patterns and contents of extracts tested for anti-adipogenic effects in literature, nor on the choice of the microalgae, but on the other hand on growth rates comparing micro- and macroalgae which does not relate to the topic of the manuscript
  • methodology cultivation of algae: it is inapprehensible, why the fresh water algae Desmodesmus armatus has been cultivated in a sea water medium (marine, F/2, not F2);
    - bioreactor operation volume should be given, as well as the aeration rate. According to the given data it appears that the later was very low in the experiments, if the numbers are correct, sedimentation/dead zones will most likely have occurred. This leads to senescence of algae and their pigments, supported by the finding of phaeophorbide a (shown in fig. 2 B and C), a circumstance that was not discussed
    - how measurement of growth/ reaching of stationary phase was carried out was not given (line 413)
    - biological terms are used wrong: Chlorophyceae is referring to the class, not the family (Linné), the term strain is also not suitable here, the authors compare different algal species out of different divisions and classes
    - line 419 (example) and throughout the manuscript: the term concentration is used synonymously for proportions, although it is defined with a reference to a volume (mass or mol)
    - extraction conditions (sonication, over night (highly inaccurate information!)) will most likely lead to decay/degradation of pigments, unsuitable for the analysis and comparison of different algae species; rather polar extraction solvent mixture may discriminate hydrocarbons, such as beta-carotene (not found in the analysis)
  • the pigment analytical part contains technical mistakes and insufficient results
    - using a wave length range until 640 nm if chlorophylls and its derivatives shall be analyzed is not useful, as the absorption maxima are found above 640 nm  (652 to 666 nm)
    - differentiation of cis- and all-E fucoxanthin is not discussed, differentiation between the 2 isomers not tracebale/documented, typical cis-band is not recorded (table 1), if a cis-peak is identified, the all-E-peak should also be termed accordingly
    - quantification of pigments is unclear, “amount %” as g/ 100 g?, area%?, normalized ion intensity? Was a calibration carried out? The statistics for the analytics are completely missing.
    - major peaks are not identified (e.g. peak 1), although MS/MS have been applied, spectral information is missing, for what reason? the identification of peak 1 as chlorophyll is not comprehensible (example)
    - the occurrence of a secondary carotenoid, hydroxycarotenone, is not discussed at all (and very unlikely to occur under the conditions used)
    - beta-carotene is completely missing, a circumstance, that is not even mentioned
    - diatoxanthin does not occur in Desmodesmus sp., not even in Chlorophyceae; the chlorophyll c, which is typical for Phaedactylum tricornutum in turn, has not been detected, nor was it’s missing noted
    - the pigment analytical discussion is very brief (about 40 lines), the scientific insight is poor, the difference in the overall pigment pattern was discussed with misleading terms (line 228)
  • none of the analyzed components (pigments and TAG) have been related to the measured effects in cell culture

The overall scientific merit appears to be low in the current form, the work appears in a pure data presentation form.

Author Response

Reviewer 1

The manuscript deals with microalgae extracts derived after cultivation of these in bubble columns and consists of two major parts, the LC-MS-analysis of the occurring carotenoids and the effects of algae extracts on cell cultures (pre-adipocytes).

The overall impression on thoroughness on the style and form of the manuscript is low (e.g. wrong numbering of figure 6, line 189 vs. line 172, y-axis of figure 2 missing), line 404: 16:18 h?; line 347: 50 µM (based on what molecule/molecular weight?)

The reviewer is right and the mistakes have been corrected in this revised version of the manuscript:

- Figures numeration is corrected.

- Figure 2: The x and y-axis units are indicated at the bottom of the figure as “mAU vs. Acquisition time (min).” This means the y-axis is expressed in milli-absorbance units (mAU).

- 50 µM has been changed for 50 µg/mL (line 386).

- 16:18 h has been changed for 16:8 h (line 457).

  • the introduction does not contain information on carotenoid patterns and contents of extracts tested for anti-adipogenic effects in literature, nor on the choice of the microalgae, but on the other hand on growth rates comparing micro- and macroalgae which does not relate to the topic of the manuscript.

In response to the reviewer’s comment, we have incorporated in the introduction section information from the literature regarding the main effects of the pigments present in the three extracts (lines 84-93). Regarding the choice of the microalgae, we have justified it in this revised version (lines 79-83).

  • methodology cultivation of algae: it is inapprehensible, why the fresh water algae Desmodesmus armatus has been cultivated in a sea water medium (marine, F/2, not F2);

The reviewer is right and this has been corrected in the revised version of the manuscript (lines 448-449).

- bioreactor operation volume should be given, as well as the aeration rate. According to the given data it appears that the later was very low in the experiments, if the numbers are correct, sedimentation/dead zones will most likely have occurred. This leads to senescence of algae and their pigments, supported by the finding of phaeophorbide a (shown in fig. 2 B and C), a circumstance that was not discussed.

We acknowledge the reviewer for his/her comment. The aeration rate low limit was 0.08vvm, which was elected due to the recommendation of the manufacturer, who says that the aeration should be between 6-8 L/min when the PBR is full. This rate is certainly below the rates recommended by some articles (Bitog et al. 2014) for avoiding the dead zones, but no very far because they recommend 0.1 vvm, and never exceed 0.15 vvm. We opted for a compromise between the manufacturer recommendation (0.06-0.08) and this low limit (0.1) for avoiding the dead zones.

  • Bitog JP, Lee IB, OH HM, Hong SW, Seo IH, Kwon KS. Optimised hydrodynamic parameters for the design of photobioreactors using computational fluid dynamics and experimental validation. Biosystems Engineering, 2014,122, 42-61.

- how measurement of growth/ reaching of stationary phase was carried out was not given (line 413)

The information has been included in the revised version of the manuscript (line 467-468).

- biological terms are used wrong: Chlorophyceae is referring to the class, not the family (Linné), the term strain is also not suitable here, the authors compare different algal species out of different divisions and classes

The reviewer is right. Corrections have been made in the revised version of the manuscript. Moreover, the first sentence of the paragraph "Tetradesmus obliquus and Desmodesmus armatus are green microalgae belonging to the Chlorophyceae, whereas Phaeodactylum tricornutum is a yellow-brown microalga from the Bacillariophyceae" has been moved to M&M section (lines 438-441) because we think that it is more appropriated.

- line 419 (example) and throughout the manuscript: the term concentration is used synonymously for proportions, although it is defined with a reference to a volume (mass or mol)

We acknowledge the reviewer for this comment. This has been corrected in the new version of the manuscript (line 476).

- extraction conditions (sonication, overnight (highly inaccurate information!)) will most likely lead to decay/degradation of pigments, unsuitable for the analysis and comparison of different algae species; rather polar extraction solvent mixture may discriminate hydrocarbons, such as beta-carotene (not found in the analysis).

According to the reviewer’s suggestion, we have completed the information regarding the extraction conditions. After extensively reviewing different methods and techniques for obtaining bioactive compounds from microalgae, we selected those most frequently reported ones in the literature. The extraction yield of each bioactive compound varies depending on the target compound; therefore, we explored methodologies aimed at maximizing the recovery of pigments, which are abundant in microalgae.

García-Vaquero et al. (2021) provide a comprehensive overview of green extraction technologies, which guided our selection of the most appropriate methods. Considering the increasing concern about pollution and laboratory waste in recent decades, we prioritized the application of commonly used “green” techniques for our extractions.

Several tests were conducted to determine the optimal solvent type and ratio for maximizing the recovery pigments, following the approaches of different authors (Matos et al., 2019; Pereira et al., 2021; Cesário et al., 2022). The solvents tested were methanol, ethanol, and water, with the following proportions: methanol or ethanol 90:10 distilled water; methanol or ethanol 70:30 distilled water; methanol or ethanol 50:50 distilled water; and pure methanol, ethanol, or water (100%). After extraction, the pigments were quantified; the mixtures containing methanol or ethanol at a 70:30 ratio with water yielded the highest recovery, and ethanol was selected over methanol due to its lower toxicity.

The extraction procedure was established by modifying the protocol of Cesário et al. (2022) for the extraction of total chlorophylls and carotenoids. Adjustments were made to the amount of biomass, solvent volume, and temperature in order to minimize the loss of thermosensitive compounds. Regarding the overnight extraction, as the reviewer pointed out, we agree that leaving the extraction mixture for extended periods may lead to pigment degradation. According to the literature, there is considerable variation in this regard: some protocols recommend only a few hours, while others extend the process up to 24 hours. Since 24 hours seemed excessive and could potentially cause degradation of pigments, we opted for an intermediate approach. Specifically, the mixtures were kept in an orbital shaker for 12 hours, thus avoiding both extremes. Therefore, in our case, the “overnight” extraction corresponded to approximately 12 hours. This fact has been specified in the revised version of the manuscript (line 484).

  • Garcia-Vaquero M, Sweeney T, O´Doherty J, Rajauria G. Recent advances in the use of greener extraction technologies for the recovery of valuable bioactive compounds from algae. Recent Advances in Micro and Macroalgal Processing: Food and Health Perspectives. 2021 John Wiley & Sons, Ltd
  • Matos J, Cardoso C, Gomes A, Campos AM, Falé P, Afonso C, Bandarra NM. Bioprospection ofIsochrysis galbanaand its potential as a nutraceutical. Food & Function, 2019, 10(11), 7333-7342.
  • Pereira S G, Teixeira-Guedes C, Souza-Matos G, Maricato É, Nunes C, Coimbra MA, Teixeira JA, Pereira RN, Rocha CM. Influence of ohmic heating in the composition of extracts from Gracilaria vermiculophylla. Algal Research, 2021, 58, 102360.
  • Cesário CDC, Soares J, Cossolin JFS, Almeida AVM, Sierra JJB, De Oliveira Leite M, Nunes MC, Serrão JE, Martins MA, Coimbra JSDR. Biochemical and morphological characterization of freshwater microalga Tetradesmus obliquus (Chlorophyta: Chlorophyceae). Protoplasma, 2021 259(4), 937-948.

  • the pigment analytical part contains technical mistakes and insufficient results

The issues related to the analytical part have been answered below.

- using a wave length range until 640 nm if chlorophylls and its derivatives shall be analyzed is not useful, as the absorption maxima are found above 640 nm  (652 to 666 nm).

The UV spectra of chlorophylls and their derivatives show two main absorption bands: one in the blue region (approximately 410–450 nm), known as the Soret band, and another in the red region (approximately 660–680 nm), the Q band, both corresponding to visible light absorption. Our DAD detector operates in a range of 190-640 nm and does not extend beyond 640 nm. For technical reasons we cannot measure the Q band, however chlorophylls were identified by the Soret band and confirmed by the pseudomolecular ion [M+H]+ accurate mass and their MSMS product ions.

- differentiation of cis- and all-E fucoxanthin is not discussed, differentiation between the 2 isomers not tracebale/documented, typical cis-band is not recorded (table 1), if a cis-peak is identified, the all-E-peak should also be termed accordingly.

Peaks 2 and 7 of the HPLC-DAD chromatographic profile of Phaeodactylum tricornutum were tentatively identified as fucoxanthin isomers. The major isomer (peak 2) was annotated as fucoxanthin (all-E fucoxanthin), whereas peak 7 was annotate as the cis-isomer. This assignment is consistent with the UV profile and the HRMS data. Although both isomers show the same MS fragmentation behaviour, a distinctive cis-band at 330nm was observed for peak 7 (See Figure 1 below).

Figure 1. UV spectra for peaks 2 (A) and 7 (B)

According to this observation, peak 2 was renamed as all-E fucoxanthin and the cis-band of peak 7 was included in the Table. Both isomers were previously reported by Barcenas-Pérez et al., 2021 in Phaeodactylum tricornutum.

  • Bárcenas-Pérez D, Střížek A, Hrouzek P, Kopecký J, Barradas M, Sierra-Ramírez A, Fernández-Marcos PJ, Cheel J. Production of Fucoxanthin from Phaeodactylum tricornutum Using High Performance Countercurrent Chromatography Retaining Its FOXO3 Nuclear Translocation-Inducing Effect. Marine Drugs, 2021,19, 517.

- quantification of pigments is unclear, “amount %” as g/ 100 g?, area%?, normalized ion intensity? Was a calibration carried out? The statistics for the analytics are completely missing.

Absolute quantification was not performed due to the lack of standards. Instead, peak area % from the LC–DAD chromatograms (at 470 nm) was reported for each pigment as an estimate of its relative abundance to comparatively evaluate the pigment profiles of the three target microalgal extracts.

We are aware that the % of area values do not accurately reflect the molar or mass proportions of each pigment.  However, at their blue–green absorption maxima (≈440–452 nm), the identified carotenoids exhibit molar absorption coefficients (ε) within a similar range. Therefore, the use of percent composition based on chromatographic peak areas is an accepted practice in the absence of standards, as these values provide a reasonable approximation for comparing similar extracts or samples with a comparable profile; as in our case. This is now clarified in the manuscript (Caption of Figure 1), and pointed out in the text (lines 530-535).

- major peaks are not identified (e.g. peak 1), although MS/MS have been applied, spectral information is missing, for what reason? the identification of peak 1 as chlorophyll is not comprehensible (example)

The UV–Vis spectral profiles of peaks 1 and 13 are consistent with chlorophylls. However, we were not able to confirm their identities by MS. Therefore, these peaks were tentatively annotated as unspecified chlorophylls.

- the occurrence of a secondary carotenoid, hydroxycarotenone, is not discussed at all (and very unlikely to occur under the conditions used)

The reviewer is right. After careful re-evaluation of the structural information from MS data, the annotation of this compound as hydroxycarotenone is not consistent with its molecular formula.

The MS/MS fragmentation data strongly support a minor lutein isomer as the most consistent structure. The ion at m/z 551.4248 matches the [M+H–H₂O]⁺ of a C₄₀H₅₆O₂ diol (calc. 551.4247), indicating in-source dehydration typical of xanthophylls. The second intense ion, m/z 429.3727, is the hallmark ε-ring (α-ionone) cleavage seen for lutein.

- beta-carotene is completely missing, a circumstance, that is not even mentioned.

We can confirm that beta-carotene was not present in the target extracts. This is now commented in the text (lines 255-265).

- diatoxanthin does not occur in Desmodesmus sp., not even in Chlorophyceae; the chlorophyll c, which is typical for Phaedactylum tricornutum in turn, has not been detected, nor was it’s missing noted

Diadinoxanthin and diatoxanthin were identified in our extracts from Phaeodactylum tricornutum. The diadinoxanthin–diatoxanthin cycle is well documented in diatoms and both carotenoids were previously reported in P. tricornutum.

Although the occurrence of diatoxanthin in Desmodesmus was previously reported by Safafar et al.(2015), we agree with the reviewer that this carotenoid is not frequently observed in Chlorophyceae. While the aforementioned reference could support our tentative identification of diatoxanthin in Desmodesmus, the absence of a distinct UV spectrum for this compound—overlapped by an intense peak of pheophorbide a—does not allow us to sustain the proposed annotation. This has now been corrected in Table 1.

In our targeted analyses of Phaeodactylum tricornutum extracts, we were unable to detect either of the two chlorophyll c types (c₁ and c₂).

  • Safafar H, van Wagenen J, Møller P, Jacobsen C. Carotenoids, Phenolic Compounds and Tocopherols Contribute to the Antioxidative Properties of Some Microalgae Species Grown on Industrial Wastewater. Marine Drugs, 2015, 13, 7339–7356.

- the pigment analytical discussion is very brief (about 40 lines), the scientific insight is poor, the difference in the overall pigment pattern was discussed with misleading terms (line 228)

According to the reviewer's comment, the pigment analysis discussion has been extended in the revised version (lines 238-275).

  • none of the analyzed components (pigments and TAG) have been related to the measured effects in cell culture

Yes, the reviewer is right. We have not related the observed effects with individual components of the extracts with the biological effects because to establish a clear relationship it is necessary to carry out cell incubation studies with each individual compound and with different bioactive compound fractions, as well as studies to asses additive, synergistic or antagonist effects. Without this information, it is not reasonable to make a strong proposal. Considering this, in the previous version of the manuscript we included a first approach of the main potential responsible compounds (lines 299-310).

The overall scientific merit appears to be low in the current form, the work appears in a pure data presentation form.

Reviewer 2 Report

Comments and Suggestions for Authors

This study evaluated the inhibitory effects of three types of microalgae (Monoraphidium sp., Phaeocystis globosa, and Armored Glaucophyte) pigment extracts on adipogenesis in 3T3-L1 preadipocytes. The results showed that all the extracts significantly reduced triglyceride accumulation by inhibiting the Pparg gene, and there were differences in the minimum effective dose and action intensity among different microalgae. The academic value lies in revealing the anti-obesity potential of microalgae pigments (such as chlorophyll and phaeocystis yellow pigment), providing new evidence for the development of anti-fatogenesis therapies based on natural products. However, the mechanism research is relatively shallow, and further verification of the effect of single components and in vivo effects is needed.   

Major Issues  

The dosage range (6.25 - 50 µg/mL) is solely based on previous studies and lacks pre-experimental data (such as IC50 determination) or dose-effect curve support. This may affect the reliability of the conclusion. 

It is hypothesized that chlorophyll or phaeoprotein is the key component, but the assumption has not been verified through single-component experiments (such as purifying chlorophyll or phaeoprotein), and it is impossible to rule out the synergistic or antagonistic effects of other components.   

Only using the 3T3-L1 cell model, without animal in vivo experiments or preclinical data, the conclusions should be treated with caution when extrapolated to physiological/pathological conditions. 

Only the expression of the Pparg gene was detected, without analyzing other key fatogenesis regulatory factors (such as C/EBPα, SREBP1c) or downstream target genes (such as aP2, AdipoQ). This cannot fully explain the action pathway. The experiment of knockout/overexpression was not conducted to verify whether Pparg is the sole key target.   

Minor Issues  

The details of the experimental methods are missing. The number of repetitions for the cell experiments (the "n" value) and the statistical methods (such as the type of ANOVA) are not clearly specified, which affects the reproducibility of the results. The specific model or brand of the commercial reagent kits are not described, which may reduce the transparency of the methods.   

The chart information is incomplete. The result mentions "the differences in the minimum effective doses of different microalgae", but no specific data (such as IC50 values) or significance markers (*p value range) are provided. 

The discussion of the logic needs to be optimized. There is a lack of direct literature support for the comparison of the effects of chlorophyll and phaeoprotein (such as citing previous studies on chlorophyll's anti-fat formation effect).  

The correlation between the experimental concentration (50 µg/mL) and the actual bioavailability in animals/humans has not been discussed.   

Format issue. Key words are repeated (such as "microalgae" appearing multiple times), and they need to be simplified. There are duplicate paragraphs in the author information and citation sections (such as the Abstract and Introduction sections being repeated), and they need to be checked.   

Other Issues  

The source of the cell line and the information on ethical review were not stated.   

Some paragraphs (such as the description of microalgae taxonomy) have a relatively weak correlation with the core conclusions. It is recommended to condense them.  

Author Response

Reviewer 2

This study evaluated the inhibitory effects of three types of microalgae (Monoraphidium sp., Phaeocystis globosa and Armored Glaucophyte) pigment extracts on adipogenesis in 3T3-L1 preadipocytes. The results showed that all the extracts significantly reduced triglyceride accumulation by inhibiting the Pparg gene, and there were differences in the minimum effective dose and action intensity among different microalgae. The academic value lies in revealing the anti-obesity potential of microalgae pigments (such as chlorophyll and phaeocystis yellow pigment), providing new evidence for the development of anti-fatogenesis therapies based on natural products. However, the mechanism research is relatively shallow, and further verification of the effect of single components and in vivo effects is needed.   

Major Issues  

The dosage range (6.25 - 50 µg/mL) is solely based on previous studies and lacks pre-experimental data (such as IC50 determination) or dose-effect curve support. This may affect the reliability of the conclusion. 

We acknowledge the reviewer for this comment. We agree with the reviewer that IC50 is an appropriate system to make a screening of bioactive compounds, mainly under a pharmacological point of view. Nevertheless, the IC50, which indicates the amount of a compound that produces 50% inhibition of a process, gives us information regarding the "potency" of a compound, but not its "efficacy" (maximun effect induced by the compound). In our case, we are interested in both parameters.

On the other hand, bioactive compounds, such as polyphenols and pigments, very frequently do not show a typical dose-response pattern. This has been reported in the literature and observed in previous experiments from our group. Thus, it is common to find compounds that induce a high effect at low doses and a lower effect at higher doses. In these not monotonic situations, the relevance and usefulness of IC50 is somehow limited. Due to these facts, we finally decided not to use IC50 for dose selection.

  • Gómez-Zorita S, Trepiana J, González-Arceo M, Aguirre L, Milton-Laskibar I, González, M, Eseberri I, Fernández-Quintela A, Portillo MP. Anti-Obesity Effects of Microalgae. International Journal Molecular Science, 2019, 21(1), 40.

It is hypothesized that chlorophyll or phaeoprotein is the key component, but the assumption has not been verified through single-component experiments (such as purifying chlorophyll or phaeoprotein), and it is impossible to rule out the synergistic or antagonistic effects of other components.   

We sincerely appreciate the reviewer’s comment and we agree that incubating cells with isolated compounds is needed to know those that are responsible for the effects induced by an extract. Thus, taking into account that this has not been carried out in the present study, in the Discussion section we have said that the proposal of chlorophyll as main compound responsible for the anti-adipogenic effect is just a hypothesis. In addition, taking into account the reviewer's comment, in this revised version we have added that the incubation with isolated compounds and with isolated compound fractions should be carried out in order to confirm this hypothesis (lines  299-310).

Nevertheless, although assuming the interest of the incubation with isolated compounds, this is not enough. Indeed, it is important to point out that, according to our own experience in previous culture studies, sometimes the effects observed with isolated compounds are not longer observed when the incubation is performed with a combination of various compounds (the situation found when we administered an extract), due to additive or synergistic effects. Moreover, in other cases, the effect of isolated compounds is no longer maintained when various compounds are present together in the incubation medium. Consequently, additional studies devoted to analysis these interactions are also needed.

Only using the 3T3-L1 cell model, without animal in vivo experiments or preclinical data, the conclusions should be treated with caution when extrapolated to physiological/pathological conditions. 

We agree with the reviewer. Indeed, this an in vitro study that provides useful information about the anti-adipogenic effect of three algae extracts, but studies carried out in animal models are absolutely needed in order to know the real possibilities of these extracts to be used in an anti-obesity context because important aspects, such as crosstalk among tissues and bioactive compound bioavailability cannot be assessed in in vitro studies. Additionally, intervention studies in humans are also needed in a final research step because remarkable differences can be found between animal models and humans.

These ideas have been included in the final conclusion in this revised version of the manuscript (lines 422-430).

Only the expression of the Pparg gene was detected, without analysing other key fatogenesis regulatory factors (such as C/EBPα, SREBP1c) or downstream target genes (such as aP2, AdipoQ). This cannot fully explain the action pathway. The experiment of knockout/overexpression was not conducted to verify whether Pparg is the sole key target. 

We agree with the reviewer that the expression of PPARg, as well as other key adipogenic factors such as C/EBPα and, SREBP1c are crucial for pre-adipocyte differentiation. For that reason, their mRNA levels were indeed measured in the present study (Figure 6). Moreover, we measured the expression of Creb1, an early inductor of the adipogenic process, and Klf5, another key regulator of the process. Finally, we agree with the reviewer that aP2 and AdipoQ are good indicators of a mature feature of adipocytes, as there are expressed at the latest stage of adipogenesis. In the present study, instead of aP2 and AdipoQ, we selected lipogenic enzymes FAS and ACC, which also are considered appropriate markers of adipocyte maturity.   

Finally, the reviewer indicates that knockout/overexpression studies have not been conducted in the present study. We totally agree that this mechanistic approach could help us to elucidate the complete mechanism of adipogenesis inhibition exerted by the extracts. Nevertheless, taking into account that PPARg is the master gene in the control of adipogenesis (playing a pivotal role in the development of mature adiposities), and considering that with the three extracts the transcription factor shows a clear significant reduction, accompanied by a decrease in triglyceride accumulation, we believe that this observation offers a good indication that the gene is involved and a first indication of causality. Moreover, similar results have been reported by other authors using a Phaeodactylum tricornutum extract or isolated pigments. This gives support to our proposal concerning the involvement of PPARg.

Nevertheless, an in line with the reviewer's comment in the previous version of the manuscript we already indicated that PPARg appeared to be primarily responsible for the observed lipid reduction. In this revised version, we have included some lines indicating that a knockout/overexpression study would have provided a more strong evidence of the involvement PPARg in the reduction of adipogenesis, and that this represents a limitation of the study (lines 413-417).

Minor Issues  

The details of the experimental methods are missing. The number of repetitions for the cell experiments (the "n" value) and the statistical methods (such as the type of ANOVA) are not clearly specified, which affects the reproducibility of the results. The specific model or brand of the commercial reagent kits are not described, which may reduce the transparency of the methods.   

We are surprised about this comment because indeed these pieces of information indeed appear at the end of the manuscript, in the Materials and Methods section, according to the journal's format.

  • Regarding the number of repetitions for cell experiments: “Each experiment was conducted in triplicate” (line 562-563).
  • The statistical information is provided in lines 623-628.
  • The commercial reagents kit are specified in lines 586, 593, 600, 610 and the references of some methodological approaches are specified in 567, 587 and 614.

The chart information is incomplete. The result mentions "the differences in the minimum effective doses of different microalgae", but no specific data (such as IC50 values) or significance markers (*p value range) are provided. 

Yes, in the results we have indicated that there are differences in the minimum effective dose of different microalgae. This is shown in the footnote of Figure 4, where it is written that whereas Dermodesmus armatus at a dose of 6.25 µg/mL induces a significant reduction in TG (p<0.05), Tetradesmus obliquus and Phaeodactylum tricornutum, at the same dose, did not induce significant reductions.

The discussion of the logic needs to be optimized. There is a lack of direct literature support for the comparison of the effects of chlorophyll and phaeoprotein (such as citing previous studies on chlorophyll's anti-fat formation effect).  

We acknowledge the reviewer for this interesting suggestion. Although in the manuscript we have included information about the anti-obesity effects of chlorophylls (lines 331-340), there is no information about its derivative compound pheophorbide. As we consider a valuable suggestion, we have included this information in the revised version of the manuscript (lines 341-350).

The correlation between the experimental concentration (50 µg/mL) and the actual bioavailability in animals/humans has not been discussed.   

We appreciate the reviewer’s comment. Although data concerning the bioavailability of some bioactive compounds present if the microalga extracts are available in the literature, it is important to note that these studies have addressed the bioavailability of individual pigments, whereas in plant or microalga extracts these compounds coexist and may interact with one another. In fact, such interactions can result in reduced bioavailability due to physical or chemical effects, competition for transporters, altered solubility, or changes in intestinal absorption. For example, lutein has been reported to reduce the absorption of β-carotene when both are administered together, most likely through competition for the same intestinal uptake mechanisms (Van der Berg and Vliet 1998). Thus, an evaluation of the extract pigments bioavailability that could give us a valuable information to correlate the experimental concentrations to those that achieve plasma and tissue after oral administration is not reported in the literature. Consequently, at the present moment, it is not possible to establish this correlation. Nonetheless, we acknowledge this, as an important aspect that merits further investigation in future research.

  • Van der Berg H, van Vliet T. Effect of simultaneous, single oral doses of β-carotene with lutein or lycopene on the β-carotene and retinyl ester responses in the triacylglycerol-rich lipoprotein fraction of men. The American Journal of Clinical Nutrition, 1998, 68(1), 82-89.

Format issue. Key words are repeated (such as "microalgae" appearing multiple times), and they need to be simplified. There are duplicate paragraphs in the author information and citation sections (such as the Abstract and Introduction sections being repeated), and they need to be checked.   

At this stage, we are not entirely sure we have fully understood his/her concern. Regarding the keywords, we have included the following ones: microalgae, Tetradesmus obliquus, Phaeodactylum tricornutum, Desmodesmus armatus, 3T3-L1, pre-adipocytes, adipogenesis, and obesity, as we believe these represent the most relevant terms to highlight the field of our work. Nevertheless, the key words are not repeated.

Concerning the possible duplicated paragraphs, we carefully reviewed the submitted version of the manuscript but we are unable to identify any repetitions. If the reviewer could kindly indicate the specific sections, we would be grateful and will correct them immediately.

Other Issues  

The source of the cell line and the information on ethical review were not stated.   

The murine 3T3-L1 cells used in the present study have been purchased from American Type Culture Collection (Manassas, VA, USA), and this was already indicated in lines 539-540. As it is an established cell line, according to the instructions for authors of the journal (https://www.mdpi.com/journal/ijms/instructions#ethics) an ethical statement is not required.

Some paragraphs (such as the description of microalgae taxonomy) have a relatively weak correlation with the core conclusions. It is recommended to condense them. 

We agree with the reviewer that the description of the pigment pattern or the taxonomy have a relatively weak correlation with the core conclusions. Nevertheless, we consider that this is a valuable information not only to characterize the used microalgae and microalgae extracts, but also to design future studies devoted to establish a strong correlation between microalgae characteristics and biological effects. On the other hand, the pigment distribution in the three microalgae seem to be key to understand its influence in the observed biological effect. In fact, other reviewer has asked to extend this part of the discussion (lines 238-576). For that reason, we hope the reviewer agrees that we have not reduced the text. 

Reviewer 3 Report

Comments and Suggestions for Authors

The manuscript by Carr-Ugarte et al. investigates the anti-adipogenic effects of pigment extracts from three microalgae on 3T3-L1 cells. The topic is interesting and relevant to the field of functional foods and obesity prevention. The study is generally well-designed, and the data support the main conclusion. However, several issues need to be addressed to improve the clarity, robustness, and impact of the manuscript before it can be considered for publication.

  1. Line 149-154 :The results state that  armatus was effective at all doses, including 6.25 μg/mL. However, the corresponding Figure 4A does not appear to show data for the 6.25 μg/mL dose of D. armatus . Please ensure all data points discussed in the text are clearly presented and statistically verified in the figures.
  2. Line 167-169:The authors state that gene expression was analyzed only at the highest dose (50 μg/mL). While this is a valid starting point, it limits the mechanistic insight.
  3. Line 192-195, Table 1:The pigment characterization is a strength of the paper. However, the data would be significantly more useful if the concentrations of the major pigments were provided in absolute values, not just relative percentages. This is crucial for comparing the potency of different extracts and for other researchers to replicate the work.
  4. Line 228-232:The discussion hypothesizes that chlorophylls might be more potent than fucoxanthin based on the differential effectiveness of the extracts. While a interesting speculation, this claim is not directly supported by the data, as the extracts are complex mixtures.
  5. Language and Style:The manuscript is generally well-written but would benefit from thorough proofreading by a native English speaker to correct minor grammatical errors and improve phrasing.
  6. The paper'sPercent match is 37%. Please reduce it to below 20%.

Author Response

Reviewer 3

The manuscript by Carr-Ugarte et al. investigates the anti-adipogenic effects of pigment extracts from three microalgae on 3T3-L1 cells. The topic is interesting and relevant to the field of functional foods and obesity prevention. The study is generally well-designed, and the data support the main conclusion. However, several issues need to be addressed to improve the clarity, robustness, and impact of the manuscript before it can be considered for publication.

  1. Line 149-154: The results state that  armatus was effective at all doses, including 6.25 μg/mL. However, the corresponding Figure 4A does not appear to show data for the 6.25 μg/mL dose of D. armatus . Please ensure all data points discussed in the text are clearly presented and statistically verified in the figures.

In order to check the reviewer concern, we have carefully revised the manuscript. However, we do not correctly understand what the reviewer is referring to. The cells treated with all the doses of Desmodesmus armatus reduced their triglyceride content, which has been expressed by a symbol in the Figure. In addition, triglyceride reduction percentages were indicated in the text (lines 156-158 of the revised version).

  1. Line 167-169: The authors state that gene expression was analyzed only at the highest dose (50 μg/mL). While this is a valid starting point, it limits the mechanistic insight.

We acknowledge the reviewer for her/his valuable comment. In fact, we agree that different criteria can be used to select the dose at which the gene expression analysis is performed. In this case, we selected each dose of extract showing the greatest triglyceride reduction that is 50 μg/mL in all cases. That way, we analysed the anti-adipogenic mechanism at the most effective dose tested in this experiment.   

  1. Line 192-195, Table 1: The pigment characterization is a strength of the paper. However, the data would be significantly more useful if the concentrations of the major pigments were provided in absolute values, not just relative percentages. This is crucial for comparing the potency of different extracts and for other researchers to replicate the work.

The reviewer is right and the absolute values would be really interesting. However absolute quantification was not performed due to the lack of standards. Instead, peak area % from the LC–DAD chromatograms (at 470 nm) was reported for each pigment as an estimate of its relative abundance to comparatively evaluate the pigment profiles of the three target microalgal extracts.

We are aware that the % of area values do not accurately reflect the molar or mass proportions of each pigment.  However, at their blue–green absorption maxima (≈440–452 nm), the identified carotenoids exhibit molar absorption coefficients (ε) within a similar range. Therefore, the use of percent composition based on chromatographic peak areas is an accepted practice in the absence of standards, as these values provide a reasonable approximation for comparing similar extracts or samples with a comparable profile; as in our case. This is now clarified in the manuscript (Caption of Figure 1), and pointed out in the text (lines 530-535).

  1. Line 228-232:The discussion hypothesizes that chlorophylls might be more potent than fucoxanthin based on the differential effectiveness of the extracts. While a interesting speculation, this claim is not directly supported by the data, as the extracts are complex mixtures.

We agree with the reviewer that working with extracts containing mixtures of bioactive compounds make difficult to speculate about the most responsible one in the observed biological effect. The reviewer is right and maybe this point could be too speculative, so this fact has been pointed out in the revised version of the manuscript (lines 309-312). 

  1. Language and Style:The manuscript is generally well-written but would benefit from thorough proofreading by a native English speaker to correct minor grammatical errors and improve phrasing.

We acknowledge the reviewer for the advice. In fact as we are not native English speakers, all of our manuscript are corrected by a native English speaker to improve the quality of our text. We hope that the new version of the manuscript fit well with the standards of the journal.

  1. The paper'sPercent match is 37%. Please reduce it to below 20%.

We have made some modifications in the revised version of the manuscript and we think that the percentage match in lower now.

Round 2

Reviewer 2 Report

Comments and Suggestions for Authors

Although the issues I raised were not fully resolved, it cannot be denied that this article still has certain academic value and can be considered for acceptance.

Reviewer 3 Report

Comments and Suggestions for Authors

no further comment.